# The Association between Burnout, Social Support, and Psychological Capital among Primary Care Providers in Togo: A Cross-Sectional Study

**DOI:** 10.3390/medicina59010175

**Published:** 2023-01-15

**Authors:** Solim Essomandan Clémence Bafei, Jiaping Chen, Yinan Qian, Lei Yuan, Yimin Zhou, Muhammed Lamin Sambou, Anita Nyarkoa Walker, Wei Li, Sijun Liu

**Affiliations:** 1Department of Social Medicine and Health Education, School of Public Health, Nanjing Medical University, Nanjing 211166, China; 2Department of Epidemiology, School of Public Health, Nanjing Medical University, Nanjing 211166, China; 3Department of Anesthesiology, Children’s Hospital of Nanjing Medical University, Nanjing 210008, China

**Keywords:** primary care providers, job burnout, social support, psychological capital

## Abstract

*Background and Objectives*: Job burnout is prevalent among primary care providers (PCPs) in different countries, and the factors that can alleviate burnout in these countries have been explored. However, no study has addressed the prevalence and the correlates of job burnout among Togolese PCPs. Therefore, we aimed to examine the prevalence of burnout and its association with social support and psychological capital among PCPs in Togo. *Material and Methods*: We conducted a cross-sectional study in Togo from 5 to 17 November 2020 among 279 PCPs of 28 peripheral care units (PCUs). Participants completed the Maslach Burnout Inventory, Job Content Questionnaire, and Psychological Capital Questionnaire. Data were analyzed using the Mann–Whitney *U* test, Kruskal–Wallis *H* test, Pearson correlation analysis, and multiple linear regression. *Results:* We received 279 responses, out of which 37.28% experienced a high level of emotional exhaustion (EE), 13.62% had a high level of depersonalization (DP), and 19.71% experienced low levels of personal accomplishment (PA). EE had a significant negative correlation with the supervisor’s support. In contrast, self-efficacy, hope, optimism, and resilience had a significant negative correlation with DP and a significant positive correlation with PA. Furthermore, supervisors’ support significantly predicted lower levels of EE. Optimism significantly predicted lower levels of DP and higher levels of PA. *Conclusions:* Burnout is common among Togolese PCPs, and self-efficacy, optimism, and supervisors’ support significantly contribute to low levels of job burnout among Togolese PCPs. This study provided insight into intervention programs to prevent burnout among PCPs in Togo.

## 1. Introduction

Primary care (PC) is defined as the basic healthcare provided to all community members at a low cost to individuals, communities, and the country [1]. Its services include illness prevention, treatment, management, rehabilitation, and palliation. PC providers (PCPs) are responsible for providing PC to individuals and the community, and in periods of health emergencies such as the COVID-19 pandemic, they provide a critical first line of defense and response to keep people safe and healthy while providing essential health services to communities [2]. A good working PC system requires a skilled, effective, and motivated workforce. However, African PCPs frequently encounter limited resources in the rural, remote practice settings in which they often deliver care [3], which might contribute to a feeling of a lack of support, disillusion, or burnout [3]. 

Burnout is a psychological condition that occurs as an extended response to chronic interpersonal occupational stressors and has three major dimensions: emotional exhaustion (EE), depersonalization (DP), and a reduced sense of personal achievement (PA) [4]. Previous literature on burnout focused on healthcare workers (HCWs) from intensive care hospitals [5,6]; nevertheless, there is increasing evidence that burnout is also prevalent among PCPs. Burnout’s prevalence among PCPs ranged from 17.30 to 50.09% in studies from Iran, Brazil, Spain, and China [1,7,8,9] and is associated with a number of negative outcomes, such as poor self-care, poor mental health, medical errors, low performance, higher turnover, and absenteeism among HCWs [10]

The theories that explain the occurrence of burnout consider the idea that demand outnumbers resources in work settings [11], highlighting the importance of organizational factors in the occurrence of burnout. The key drivers of burnout include occupation [12], low wages, heavy workloads, lack of support [6], and working hours [13]. The COVID-19 pandemic has kept the health systems of countries on alert all around the world and has substantially impacted the work environment. It has contributed to an increase in workload with inadequate resources, further compromising HCWs’ coping methods [14]. 

The buffering effect of social support and psychological capital on burnout has been investigated [15,16,17]. Support from colleagues and supervisors strongly influences employees’ job performance. In addition, high levels of social support result in positive health outcomes by acting as a stress resilience factor [18]. A study in Ecuador showed that social support mediated the adverse impact of burnout on health among 1035 HCWs [16]. Similarly, a study in China found that social support moderated the relationship between burnout and anxiety symptoms among 514 HCWs [6]. In a study from Hawaii, support from colleagues and supervisors contributed to lower burnout among 170 HCWs [15].

Psychological capital is an individual positive psychological state, including self-efficacy, optimism, hope, and resilience [19], which substantially affects peoples’ attitudes and behavior [20]. Thus, people with high psychological capital could be less affected by the traumatic situations of life. A study in China found that psychological capital was related to lower levels of burnout and a higher creative tendency among 200 nurses in China [17]. Similarly, a study from Iran found that psychological capital contributed to lower levels of job burnout and higher levels of mental health among nurses [21]. Psychological capital contributed to higher levels of organizational commitment and lower levels of burnout and anxiety in a study among 1354 Chinese nurses [22].

Previous studies that investigated the prevalence of burnout and its associated factors among PCPs from other countries displayed various results. However, little is known about burnout prevalence and its correlates among PCPs in Togo, particularly during an epidemic or a pandemic. Therefore, this study aimed to investigate the prevalence of burnout and its association with social support and psychological capital among PCPs (HCWs working in peripheral care units (PCUs) from two districts of Togo) during the COVID-19 pandemic.

## 2. Materials and Methods

### 2.1. Study Settings

Togo is a French-speaking, West African country with a surface area of 56,600 km^2^ and a population of 7.71 million inhabitants in 2020 [23]. Togo is bordered in the north by Burkina-Faso, in the south by the Atlantic Ocean, in the west by Ghana, and in the east by Benin. The healthcare system of the country is organized into three levels [24]:−The central level is made up of the Minister’s office, the general secretariat, and the general and central directorates; −The intermediary level, constituting six health regions;−The peripheral level, based on forty-four health districts that provide first contact care structured around three levels, namely: (i) the community health agents who by delegation provide care at the family and community level and who are called upon to play the role of interface between the community and the health services; (ii) the PCU as the basis of the healthcare system and from which PC activities are carried out in a fixed strategy and towards the populations; and (iii) the district hospital, which constitutes the first referral level.

### 2.2. Study Design 

A cross-sectional study was conducted in Togo from 5 to 17 November 2020, in the early stage of the COVID-19 pandemic. The first case of COVID-19 was reported on 6 March 2020 in Togo, and by November 2020, there were 2974 total cases with a total death count of 64 [25].

We applied multistage sampling to select participants from the six health regions of Togo. In the first stage of sampling, we selected the two most populated regions of Togo, which are Lomé-commune (two health districts and 2,002,670 inhabitants) and Plateaux (twelve health districts and 1,723,232 inhabitants). In the second stage, Agoe-Nyve and Haho districts were selected from the Lomé-commune and Plateaux regions, respectively. Figure 1 shows the sampling technique. Participants were included in our study based on the following criteria: (1) being an HCW working in a PCU of the selected districts and (2) consenting to participate in this survey. In total, 292 PCPs were invited to participate in this study, and 279 people aged 22–59 years (112 males, 167 females) completed the survey. The study targeted PCPs, including physician assistants, nurses, midwives, laboratory workers, hygiene specialists, and nursing aids.

### 2.3. Procedure

The data were collected using an online survey with the aid of the District Health Directorates. In the Agoe-Nyve district, the district health directorate gave information about the study objectives to healthcare facilities leaders; afterward, the leader of each health facility was in charge of forwarding the study participation invitation involving the study objectives and the link hosting the survey in their health facilities’ WhatsApp groups that involved the PCPs of that district. In the Haho district, the district health directorate charged one person to forward the link hosting the survey to the district’s WhatsApp group that involved the PCPs working in that district. The recruitment procedure was anonymous and voluntary, and completing that survey implied consent. To ensure respondents’ anonymity, information that can allow their identification was not collected. The Togolese Ministry of Health approved the study.

### 2.4. Measured Variables

The survey questionnaire contains four sections:

The first section covered demographic and work-related characteristics such as gender, age, marital status, education qualification, occupation, work location, number of working years’ experience, shift, number of hours worked in a typical week, and number of patients attended in a typical day.

The second section covered the Maslach Burnout Inventory (MBI), which includes EE (number of items (N) = 9), DP (N = 5), and PA (N = 8). The questionnaire comprised a seven-point Likert scale, ranging from 0 to 6 (0 = never, and 1 to 6 represent daily experience levels), representing burnout levels experienced daily [26]. According to MBI, higher EE (≥30) and DP (≥12) scores and a lower PA (≤33) score indicate a high level of burnout. MBI is widely used, translated, and validated in French [4,27]. The coefficient alpha internal consistency of scale calculated in this study was 0.86.

The third section covered social support at the workplace, measured on a four-point Likert scale (1 = strongly disagree to 4 = strongly agree) using the Job Content Questionnaire (JCQ) developed by Karasek et al. [28]. It contains eight items: four for supervisors’ support and four for colleagues’ support. The JCQ is widely used, translated, and validated in French [27,29]. The coefficient alpha internal consistency of scale calculated in this study was 0.79.

The fourth section covered psychological capital, measured using the Psychological Capital Questionnaire (PCQ) [30]. It contains 24 items related to self-efficacy (6 items), hope (6 items), resilience (6 items), and optimism (6 items). Psychological capital was measured on a six-point Likert scale (1 = strongly disagree to 6 = strongly agree). The PCQ is widely used, translated, and validated in French [31,32]. The coefficient alpha internal consistency of PCQ was 0.85 in this study.

### 2.5. Data Analysis 

Continuous variables were texted for normality, frequencies, and medians, and interquartile ranges (IQR) were computed for qualitative and continuous non-normally distributed data, respectively. Mann–Whitney U and Kruskal–Wallis H tests were used to determine the significant difference among groups according to burnout dimensions for non-normally distributed data. Pearson correlation analysis was performed to determine the correlations between burnout dimensions, social support, and psychological capital. Furthermore, we performed multiple linear regression analyses to describe the associated factors. Each burnout dimension was modeled separately against the independent variables in the multiple linear regression analysis. Only those independent variables with *p* < 0.05 in univariate analysis were considered in the regression analysis. All statistical analyses were performed in SPSS software version 26.0, and statistical significance was set at *p* < 0.05.

## 3. Results

Out of 292 survey invitations sent out, 279 PCPs took part in the survey, giving a response rate of 95.55%. About 37.28%, 13.62%, and 19.71% of the study sample exhibited a high level of EE and DP and a low level of PA, respectively.

### 3.1. Baseline Characteristics of Participants According to EE, DP, and PA

Table 1 shows the baseline characteristics of the study sample and the distribution of burnout. We found that 59.86% of the participants were female, 56.63% were aged 36 and older, 77.42% were married, and 54.48% had a bachelor’s degree. We have also found that 59.50% were nurses and midwives; 74.19% worked in the Agoe-Nyve district, and 41.58% worked for 11 years or more. About 61.29% of the participants worked rotating shifts; 63.44% worked 41 h per week and above, and 64.52% attended to at least 16 patients daily. Furthermore, according to the burnout dimensions (EE, DP, and PA), a significant difference in EE scores among characteristics such as gender (*U* = 10750.00; *p* = 0.034), work location (*U* = 11360.50; *p* < 0.001), and number of patients attended daily (*U* = 11808.00; *p* < 0.001) was observed. There were significant differences in DP scores among subgroups such as gender (*U* = 10750.50; *p* = 0.032), marital status (*U* = 8336.50; *p* = 0.006), age (*U* = 7787.00; *p* = 0.007), work location (*U* = 9837.00; *p* < 0.001), and the number of patients attended on a typical day (*U*=10681.00; *p* = 0.005). In addition, a significant difference in PA scores was observed for gender (*U* = 7889.50; *p* = 0.027).

### 3.2. Correlations Analysis of Burnout, Social Support, and Psychological Capital

Table 2 represents the correlation of job burnout dimensions, social support, and psychological capital variables. There was a significant negative correlation between EE and supervisors’ support (r = −0.186; *p* = 0.002). DP negatively correlated with self-efficacy (*r* = −0.268; *p* < 0.001), hope (*r* = −0.170; *p* = 0.004), optimism (*r* = −0.205; *p* = 0.001), and resilience (*r* = −0.164; *p* = 0.006). PA positively correlated with self-efficacy (*r* = 0.260; *p* < 0.001), hope (*r* = 0.145; *p* = 0.015), optimism (*r* = 0.211; *p* < 0.001), and resilience (*r* = 0.136; *p* = 0.023).

### 3.3. Multiple Linear Regression Analysis

Table 3 shows the results of the multiple linear regression analysis. Working at Agoe-Nyve district (*β* = 0.320; *p* < 0.001) and attending to 16 patients and above daily (*β* = 0.120; *p* = 0.046) were positively associated with EE, while supervisors’ support was negatively related to EE (*β* = −0.148; *p* = 0.007). Working at Agoe-Nyve district (*β* = 0.177; *p* = 0.006) was associated with higher DP; self-efficacy (*β* = −0.155; *p* = 0.031) and optimism (*β* = −0.149; *p* = 0.019) were related to lower DP and higher PA (*β* = 0.230; *p* = 0.001 and β = 0.149; *p* = 0.021, respectively). 

## 4. Discussion

This study is the first to explore the prevalence of burnout and its association with social support and psychological capital among PCPs in Togo. About 37.28% of the surveyed PCPs had high levels of EE, 13.62% had high levels of DP, and 19.71% had low levels of PA. 

The prevalence of EE in our study was higher than in the findings of a recent meta-analysis of 31 studies among PCPs from low- and middle-income countries [33] and a cohort study among PCPs from Ethiopia [34]. PCPs serve as the first contact of individuals in the healthcare system and provide essential medical and public health services to individuals and communities. Moreover, the mandate of Togolese PCPs is not only limited to the PCU, but also includes activities such as vaccination, children’s nutritional status control, health education, and health promotion in hard-to-reach communities. They are also crucial in the epidemiological surveillance of diseases at primary levels and are involved in the response when there is a health emergency. All these bring about heavy workloads and long working hours which might result in higher levels of EE.

DP is a response to negative and stressful situations in daily life [35,36]. In this study, DP prevalence was lower than in the findings of a study among PCPs from Brazil [9]. A possible explanation is that our study respondents are less at risk of stressful situations in their work than others. However, the prevalence of DP was higher than the report of the study among PCPs from China [37]. In our study, the prevalence of low PA was lower than the findings of previous studies among PCPs from China [8] and Iran [38]. Regarding their proximity to the population and all the healthcare services delivered by PCPs, communities value PCPs and even consider them leaders, which could increase their feeling of PA. 

In agreement with a previous study among healthcare workers in Hawaii [15], our study showed that supervisors’ support was correlated with lower levels of EE. Similarly, a study in Poland found that supervisors’ support was negatively associated with EE and DP among nurses and midwives [39]. In this study, supervisors’ support was recognized as a significant factor in emotional exhaustion reduction. Supervisors can reduce job demands and workload, which is closely related to emotional exhaustion [4]. In times of crisis, such as the COVID-19 pandemic, supervisors’ roles are more essential because of their abilities in sustaining subordinates’ work morale and psychological well-being [40]. For Togolese PCPs, the immediate supervisors’ support comes from PCUs management. Intervention programs to enhance supervisory skills and relationships with subordinates could contribute to reducing EE levels, which will reduce burnout. 

Our study has also assessed the relative contribution of the underlying components of psychological capital on burnout. We found that self-efficacy, hope, optimism, and resilience had a significant negative correlation with DP and a significant positive correlation with PA, which is consistent with previously published literature [41]. In the multiple linear regression models, we found that optimism significantly predicted lower levels of DP, while optimism and self-efficacy significantly predicted higher levels of PA. Likewise, a study in China found that hope and optimism were associated with lower levels of EE while optimism was associated with a lower level of DP, and self-efficacy and optimism were associated with higher levels of PA [42]. A recent study among nurses from Turkey also reported that higher self-efficacy and optimism contributed to lower levels of burnout, explaining 26% of burnout variance [43]. Studies that used psychological capital as a combined construct have also found that higher psychological capital contributed to lower levels of burnout [17,21]. All these show that psychological capital is an important factor in the explanatory model of burnout. During health emergencies such as the COVID-19 pandemic, HCWs with higher psychological capital might cope better with the changes in their work environments. Intervention programs such as psychological capital development training, positive psychology training interventions, or reading interventions to improve psychological capital, especially self-efficacy and optimism, will effectively slow down or eliminate burnout among PCPs. 

The effect of work-related characteristics on the three dimensions of burnout was also examined. The PCPs working in the Agoe-Nyive district were more likely to experience EE and DP than their counterparts in the Haho district. The demands for healthcare services in urban areas are higher than in rural areas. However, this result contradicts the report of a study in the USA, which reported that study participants from rural areas experienced higher levels of EE than their counterparts in suburban and urban areas [44]. Furthermore, this relationship can also be explained by the fact that the prevalence of COVID-19 might be different in the two regions. Attending over 16 patients daily was associated with higher levels of EE, which was consistent with the results of a recent systematic review of 16 studies from the United States of America [45]. In that review, it was found that workload was a primary stressor in the work environment, and a higher workload was associated with burnout occurrence among nurses [45]. All of these results suggest that reducing working hours and workload and improving working conditions could contribute to burnout reduction or prevention. 

### 4.1. Practical Implications

Burnout contributes to poor self-care and mental health, medical errors, low performance, higher turnover, and absenteeism [8]. These could affect not only workers’ well-being but also health consumers’ safety and healthcare services’ productivity. Therefore, investigating the prevalence and the associated factors of burnout among the PCPs of the surveyed districts has provided information on their burnout status, risk, and protective factors. Reducing workload and improving working conditions as well as enhancing self-efficacy, optimism, and supervisors’ support could be part of intervention programs for burnout prevention or reduction for the study sample. Preventing or reducing burnout among Togolese PCPs could enhance PCPs’ well-being and promote their retention, which in turn might increase patients’ safety and satisfaction with healthcare services and the performance of the overall health system.

### 4.2. Strengths and Limitations 

To the best of our knowledge, this is the first study to explore the prevalence of job burnout and its relationship with social support in the workplace and psychological capital among Togolese PCPs, which can provide suggestions about intervention programs to prevent and reduce burnout in the surveyed districts. 

However, this study might be subject to limitations. First, there was a potential risk of recall bias in reporting events. Second, data were collected through an online survey, which will lead to response bias. Third, we cannot infer causality because of the cross-sectional nature of the study; although cross-sectional design can provide valuable results, it can only give information about possible risk and protective risk factors. Therefore, further longitudinal or experimental studies should be conducted to confirm the conclusion of this study. Fourth, the study was conducted in two health districts (out of the forty-four health districts in Togo) with a small sample size; thus, the results may not represent the overall national setting. Therefore, future studies at the national level are needed to investigate the burnout status of Togolese PCPs.

## 5. Conclusions

This study’s findings demonstrated that job burnout is common among Togolese PCPs. Supervisors’ support significantly predicted lower levels of EE, while optimism significantly predicted lower levels of DP and self-efficacy, and optimism significantly contributed to higher levels of PA. Leaders in the Togolese health system are called on to reduce workloads and improve working conditions, as well as strengthen PCPs’ psychological capital and enhance supervisors’ supervisory skills and interpersonal relationships by designing effective strategies to prevent burnout. This study provides new insight, which can be used in intervention programs to prevent burnout among PCPs of Togo.

## Figures and Tables

**Figure 1 medicina-59-00175-f001:**
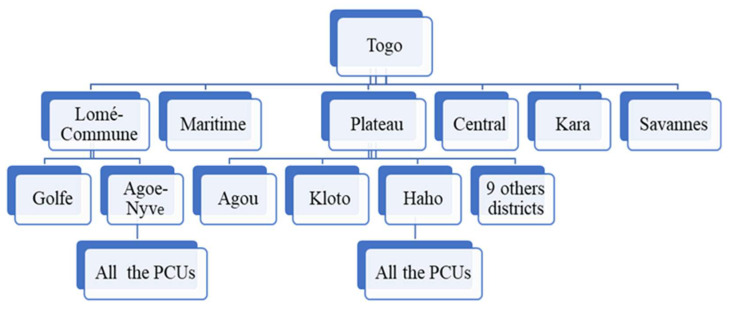
Sampling technique.

**Table 1 medicina-59-00175-t001:** Baseline characteristics of participants according to EE, DP, and PA.

Variables	*n (%)*	EE *Median (IQR)*	*p* *U/H*	DP *Median (IQR)*	*p* *U/H*	PA *Median (IQR)*	*p* *U/H*
**Gender**							
Female	167 (59.86)	26.00 (15.00–35.00)	0.034	5.00 (0.00–7.00)	0.032	41.00 (35.00–44.00)	0.027
Male	112 (40.14)	22.00 (11.25–33.00)	10,750.00	2.00 (0.00–6.00)	10,750.50	42.00 (37.00–35.00)	7889.50
**Marital status**							
Single	63 (22.58)	20.00 (14.00–32.00)	0.237	5.00 (1.00–9.00)	0.006	40.00 (35.00–45.00)	0.633
Married	216 (77.42)	26.00 (14.00–34.00)	6138.00	2.00 (0.00–6.00)	8336.50	41.50 (36.00–45.00)	6535.50
**Age (years)**							
≤35	121 (43.37)	23.00 (13.00–32.00)	0.139	5.00 (1.00–8.50)	0.007	41.00 (35.00–45.00)	0.703
≥36	158 (56.63)	26.00 (14.00–35.25)	10,547.00	2.00 (0.00–6.00)	7787.00	41.50 (36.00–45.00)	9813.50
**Occupation**							
Physician assistants	26 (9.32)	25.00 (13.00–34.00)	0.098	2.00 (0.00–12.00)	0.137	41.00 (35.25–44.00)	0.881
Nurses & midwives	166 (59.50)	26.00 (14.00–36.25)	7.842	4.00 (0.00–6.25)	6.986	41.00 (35.75–45.00)	1.183
Lab workers	25 (8.96)	19.00 (7.50–29.50)		1.00 (0.00–6.00)		42.00 (37.50–47.00)	
Hygiene specialists	22 (7.89)	28.00 (12.50–36.25)		5.50 (1.75–11.00)		40.50 (36.75–45.25)	
Nursing aides	40 (14.34)	21.50 (9.50–28.75)		1.00 (0.00–5.75)		41.50 (32.25–45.00)	
**Education qualification**							
Certificate	67 (24.01)	25.00 (12.00–32.00)	0.416	2.00 (0.00–6.00)	0.605	41.00 (33.00–45.00)	0.763
Diploma	60 (21.51)	28.50 (13.25–37.00)	1.756	3.00 (0.00–6.00)	1.006	41.00 (35.00–45.00)	0.540
Bachelor and above	152 (54.48)	24.00 (14.00–34.00)		3.00 (0.00–8.00)		41.00 (37.00–45.00)	
**Work location**							
Haho	72 (25.81)	11.50 (7.00–24.75)	<0.001	1.00 (0.00–3.00)	<0.001	42.50 (16.00–47.00)	0.936
Agoe-Nyve	207 (74.19)	28.00 (18.00–37.00)	11,360.50	5.00 (0.00–8.00)	9837.00	41.00 (36.00–44.00)	7499.00
**Working experience (years)**							
≤5	81 (29.03)	20.00 (13.00–36.00)	0.427	5.00 (1.00–9.00)	0.226	41.00 (36.50–44.50)	0.896
6–10	82 (29.39)	24.00 (12.50–33.00)	1.702	3.00 (0.00–6.00)	2.971	41.00 (35.75–45.25)	0.220
≥11	116 (41.58)	26.50 (15.25–34.00)		2.50 (0.00–6.00)		41.00 (35.00–45.00)	
**Shift worked**							
Day shift	108 (38.71)	26.00 (14.00–34.00)	0.661	3.00 (0.00–6.00)	0.830	41.00 (36.00–44.00)	0.447
Rotating shift	171 (61.29)	24.00 (13.00–34.00)	8946.00	3.00 (0.00–7.00)	9372.50	41.00 (35.00–45.00)	9732.50
**Hours worked in a typical week**							
≤40 h	102 (36.56)	26.50 (15.00–36.00)	0.133	2.00 (0.00–6.00)	0.075	42.00 (37.00–45.00)	0.103
≥41 h	177 (63.44)	24.00 (12.50–34.00)	8053.50	3.00 (1.00–6.50)	10,163.50	41.00 (35.00–45.00)	7969.50
**Patients attended daily**							
≤15	99 (35.48)	17.00 (8.00–30.00)	<0.001	1.00 (0.00–5.00)	0.005	41.00 (33.00–46.00)	0.564
≥16	180 (64.52)	27.00 (17.25–37.00)	11,808.00	5.00 (0.00–8.00)	10,681.00	41.00 (37.00–45.00)	9281.50

*Note: n* = number of participants; % = percentages; *IQR* = interquartile range; EE = emotional exhaustion; DP = depersonalization; PA = personal accomplishment.

**Table 2 medicina-59-00175-t002:** Pearson’s correlations between burnout dimensions, social support, and psychological capital variables.

Variables		1	2	3	4	5	6	7	8	9
1. EE	*r*	1	0.452	0.359	−0.186	0.074	−0.080	−0.019	−0.052	−0.117
	*p*		<0.001	<0.001	0.002	0.217	0.183	0.752	0.391	0.051
2. DP	*r*	0.452	1	0.027	−0.104	−0.043	−0.268	−0.170	−0.205	−0.164
	*p*	<0.001		0.649	0.084	0.473	<0.001	0.004	0.001	0.006
3. PA	*r*	0.359	0.027	1	0.034	0.103	0.260	0.145	0.211	0.136
	*p*	<0.001	0.649		0.572	0.085	<0.001	0.015	<0.001	0.023
4. Supervisors’ support	*r*	−0.186	−0.104	0.034	1	0.365	0.142	0.339	0.130	0.101
	*p*	0.002	0.084	0.572		<0.001	0.018	<0.001	0.030	0.092
5. Colleagues’ support	*r*	0.074	−0.043	0.103	0.365	1	0.141	0.394	0.189	0.212
	*p*	0.217	0.473	0.085	<0.001		0.019	<0.001	0.002	<0.001
6. Self-efficacy	*r*	−0.080	−0.268	0.260	0.142	0.141	1	0.524	0.318	0.410
	*p*	0.183	<0.001	<0.001	0.018	0.019		<0.001	<0.001	<0.001
7. Hope	*r*	−0.019	−0.170	0.145	0.339	0.394	0.524	1	0.374	0.549
	*p*	0.752	0.004	0.015	<0.001	<0.001	<0.001		<0.001	<0.001
8. Optimism	*r*	−0.052	−0.205	0.211	0.130	0.189	0.318	0.374	1	0.379
	*p*	0.391	0.001	<0.001	0.030	0.002	<0.001	<0.001		<0.001
9. Resilience	*r*	−0.117	−0.164	0.136	0.101	0.212	0.410	0.549	0.379	1
	*p*	0.051	0.006	0.023	0.092	<0.001	<0.001	<0.001	<0.001	

Note: EE = emotional exhaustion; DP = depersonalization; PA = personal accomplishment.

**Table 3 medicina-59-00175-t003:** Multiple linear analysis of the influencing factors of burnout.

Dependents Variables	Covariates	B	SE	β	t	*p*	95% CI for B
EE	Constant term	22.789	3.787		6.017	<0.001	15.333 to 30.246
	**Sex:** Female	1.897	1.502	0.070	1.263	0.208	−1.06 to 4.855
	District: Agoe-Nyve	9.707	1.837	0.320	5.284	<0.001	6.091 to 13.323
	Patients attended daily ≥ 16	3.336	1.663	0.120	2.006	0.046	0.063 to 6.610
	Supervisor’s support	−0.718	0.265	−0.148	−2.706	0.007	−1.24 to −0.196
DP	Constant term	15.072	3.006		4.915	<0.001	9.034 to 21.109
	Sex: Female	0.221	0.725	0.018	0.304	0.761	−1.207 to 1.648
	Age: ≥ 36	−0.637	0.726	−0.053	−0.878	0.381	−2.066 to 0.792
	Marital status: Single	1.382	0.877	0.097	1.576	0.116	−0.345 to 3.109
	District: Agoe-Nyve	2.401	0.871	0.177	2.757	0.006	0.687 to 4.115
	Patients attended daily:16	0.431	0.773	0.035	0.558	0.578	−1.091 to 1.953
	Self-efficacy	−0.937	0.431	−0.155	−2.174	0.031	−1.785 to −0.089
	Hope	−0.138	0.550	−0.019	−0.251	0.802	−1.221 to 0.945
	Optimism	−1.228	0.522	−0.149	−2.353	0.019	−2.256 to −0.201
	Resilience	−0.133	0.532	−0.018	−0.250	0.803	−1.181 to 0.915
PA	Constant term	19.167	4.695		4.083	<0.001	9.925 to 28.410
	Sex: Female	0.248	1.193	0.012	0.208	0.836	−2.101 to 2.596
	Self-efficacy	2.300	0.710	0.230	3.242	0.001	0.903 to 3.697
	Hope	−0.408	0.921	−0.034	−0.443	0.658	−2.221 to 1.405
	Optimism	2.027	0.876	0.149	2.314	0.021	0.302 to 3.752
	Resilience	0.069	0.890	0.006	0.077	0.939	−1.683 to 1.820

EE: *F* = 16.471, *p* < 0.001, *R*² = 0.194, Adjusted *R*² = 0.182; DP: *F* = 4.949, *p* < 0.001, *R*² = 0.142, Adjusted *R*² = 0.113; PA: *F* = 5.171, *p* < 0.001, *R*² = 0.087, Adjusted *R*² = 0.070; *β* = Beta standardized coefficient; *CI* = confidence interval; *B* = unstandardized coefficient effect.

## Data Availability

All data are available from the corresponding author (sjliu@njmu.edu.cn).

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
