# Peer review of "The Association between Burnout, Social Support, and Psychological Capital among Primary Care Providers in Togo: A Cross-Sectional Study"

_medicina, 2023, doi:10.3390/medicina59010175_

Round 1

Reviewer 1 Report

This is an interesting and timely paper on an important topic. More research on healthcare workforce well-being is needed, and this paper contributes useful findings in this area.

I have minor suggestions some questions for the authors which I feel, if answered in the text, would strengthen the paper:

- It would be clarifying to know why these two areas included in the study were selected rather than others, and why a larger sample was not considered (there could be many good reasons, so this is not a critique of the sample size but a desire to understand the sample selection criteria). It is a limitation of the paper when framed as a study about (all) Togolese PCPs, and a stronger justification of the site selection choice might ameliorate this.

- The paper makes scant reference to the impact of the COVID-19 pandemic; it would help non-Togolese readers to have more detailed description of the impact of the pandemic on the participating workers' workloads in 2020 - data were collected in November of 2020, which in many but not all parts of the world was an exceptionally challenging time for healthcare delivery. What was the COVID-19 mortality, prevalence, and/or hospitalization rate, in Togo at that time? 

- Somewhat related to both of the above points, the findings show a different in all the burnout outcomes of interest (EE, DP, and PA) between the two workplace regions, and the discussion suggests that this may be due to the difference in rural/urban context and the greater demands of care in urban settings. Was there a difference in COVID-19 prevalence at that time in the two regions? Were there different local policies in effect that might have impacted healthcare workers sense of control and/or self-efficacy to address the pandemic, or other workplace factors other than the urban-rural difference? While the authors probably do not have data that would affect their results, it would be interesting and helpful to hear more about the context in both the introduction and the discussion.  It is notable, for example, that hygiene specialists had the highest level of EE - could this be due to the extra demands of the pandemic?

- A strength of the study is its focus on many categories of healthcare workers, including more than physicians and/or nurses. The paper also makes appropriate use of literature grounded in the West African context. Recognizing those strengths, the authors might still consider and comment on the findings of a recent review, on burnout among nurses in the United States, by Zangaro and colleagues (published in Nursing Clinics of North America, 2022 Vol 57:1). There may be similarities and differences in the findings that the authors could discuss to enrich the discussion; burnout in the United States occurs in a setting that in many ways is very different, yet rates of burn-out are fairly comparable to those found in this review. 

- Finally, I agree with the conclusion that improving individual worker's psychological capital and the support of their supervisors is likely to help alleviate burnout, but what about reduced work hours and patient load? These were found to be just as correlated with negative outcomes, and yet they are not mentioned in the implications of the study or the conclusions. The paper should not focus only on individual skill-building as an area for future intervention but acknowledge the need for organizational or systemic changes that are equally linked to negative outcomes.

Reviewer 2 Report

Bafei et al have submitted an interesting manuscript examining the rates of burnout and contributing factors using a WhatsApp-based survey of Tongolese PCPs. This is an important topic without robust data in this population or setting, but there are a number of significant concerns with the manuscript in its current form.

Major Comments

Burnout is a common problem in healthcare today, and primary care specialties certainly have a significant risk of this condition. There is considerable literature examining the rates of burnout across physician medical specialties (not just critical care, as the authors mention in the introduction; https://pubmed.ncbi.nlm.nih.gov/32614425/ is just one of many examples), and the risk factors, which are many. An important strength of this study is its examination of a wide variety of non-physician healthcare professions, which has not been well described in the medical literature.

One of the major messages that experts in the field highlight is that the driving factors – and solutions – for burnout belong at the organization/system level, with subsequent positive impacts on the work unit and individual well-being. Unfortunately, this is a missed opportunity in the introduction, which paints a picture of the challenging situation in Togo based on a single reference in somewhat inflammatory terms and then dedicates the majority of this relatively long section to focus on individual risk factors and social settings.

The description of the study design and sampling for someone who is not very familiar with the geography of Togo is confusing, and perhaps could be clarified with 1-2 sentences that explain the organization of the healthcare system (and the significance of regions and districts) in the country since the associated figure does not seem to suggest the broad-based sampling strategy that is described. The survey description and distribution method are clear, but whether the survey was translated (and if so, the internal content validation process) and how inclusive the healthcare provider membership is within the health facilities' WhatsApp groups (and therefore the survey distribution strategy) is unclear. Although the study was approved through the Ministry of Health, there is no discussion of how data were deidentified or subjects protected for providing responses in what would generally be considered a sensitive topic. The response rate was remarkably high, raising the question as to whether any incentives were offered or if the use of “official” channels might have increased participation – and also potentially impacted the results.

The results section is quite clearly presented, although the number of significant figures included in most of the numbers in Table 1 is unnecessary and should be revised. The correlations are relatively intuitive and consistent with prior published literature, which should be reflected in the discussion.

The authors compare their results in the (quite long) discussion to a variety of studies in different geographic locations and practice settings, which I found quite confusing. Using comparative data from other primary care settings in Africa would be ideal, and if these data are not available primary care settings in other geographic areas would be a much more effective comparison. Although the recommendations for better supervisor training in light of the negative impact of bad supervisors on burnout indices are reasonable, I would suggest that reducing work hours, patient volumes, and improving working conditions is essential before any individual resilience training will take effect – and what would be necessary to accomplish these meaningful changes in Tongo is notably absent from the discussion (despite it being undoubtedly the crux of the solution).  There is also no discussion of potential limitations of this study design and survey-based data collection, including potential confounding factors and bias.

Minor Comments

In the medical literature “burnout” is commonly used as a general term, rather than using the phrase “job burnout”. I would consider deleting “job” throughout the manuscript.

“Primary care (PC)” is also a more commonly used term than “primary healthcare (PHC)”. It would be useful to clearly define who does primary care in Tongo in the introduction and methods, so as to better understand the subject population you sampled.

Introduction

Line 40  “Under-resourced” is very much a value judgment – I would delete this and state “African PCPs frequently encounter limited resources in the rural, remote practice settings in which they often deliver care”

Line 42  “Ineffectual” is another very value-laden word, and the critique of supervision and leadership is unreferenced. Consider rephrasing or deleting.
